# Characterization of Crystalline Phase of TiO_2_ Nanocrystals, Cytotoxicity and Cell Internalization Analysis on Human Adipose Tissue-Derived Mesenchymal Stem Cells

**DOI:** 10.3390/ma13184071

**Published:** 2020-09-14

**Authors:** Cristiane Angélico Duarte, Luiz Ricardo Goulart, Letícia de Souza Castro Filice, Isabela Lemos de Lima, Esther Campos-Fernández, Noelio Oliveira Dantas, Anielle Christine Almeida Silva, Milena Botelho Pereira Soares, Ricardo Ribeiro dos Santos, Carine Machado Azevedo Cardoso, Luciana Souza de Aragão França, Vinícius Pinto Costa Rocha, Ana Rosa Lopes Pereira Ribeiro, Geronimo Perez, Loyna Nobile Carvalho, Vivian Alonso-Goulart

**Affiliations:** 1Laboratory of Nanobiotechnology, Institute of Biotechnology (IBTEC), Federal University of Uberlandia, Uberlandia 38.400-000, Brazil; cris.angelico@terra.com.br (C.A.D.); goulartlr@gmail.com (L.R.G.); leticiafilice@ufu.br (L.d.S.C.F.); isabela.lemos@hotmail.com (I.L.d.L.); esther.fernandez@ufu.br (E.C.-F.); loyna.nobile@gmail.com (L.N.C.); 2Laboratory of New Nanostructured and Functional Materials, Physics Institute, Federal University of Alagoas, Maceió 57073-704, Brazil; noelio@fis.ufal.br (N.O.D.); aniellechristineas@gmail.com (A.C.A.S.); 3Laboratory of Tissue Engineering and Immunopharmacology (LETI), Oswaldo Cruz Fundation, Salvador 40.000-001, Brazil; milenabpsoares@gmail.com (M.B.P.S.); ricardoribeiro1941@gmail.com (R.R.d.S.); cariazevedo@hotmail.com (C.M.A.C.); luaragao@gmail.com (L.S.d.A.F.); vinyrocha@hotmail.com (V.P.C.R.); 4Center of Biotechnology and Cell Therapy (CBTC), Sao Rafael Hospital, Salvador 40.000-001, Brazil; 5National Institute of Metrology, Quality and Technology (INMETRO), Institute of Biomedical Engineering, Rio de Janeiro 20.000-000, Brazil; analopes0781@gmail.com; 6Brazilian Branch of Institute of Biomaterials, Tribocorrosion and Nanomedicine (IBTN), Rio de Janeiro 20.000-000, Brazil; 7Division of Metrology of Materials, National Institute of Metrology, Quality and Technology (INMETRO), Rio de Janeiro 20.000-000, Brazil; perezgeronimo@hotmail.com; 8Laboratory of Nanotechnology, Federal University of Rio de Janeiro (UFRJ), Rio de Janeiro 20.000-000, Brazil

**Keywords:** TiO_2_ nanocrystal, crystalline phase, adipose tissue stem cell

## Abstract

Titanium dioxide (TiO_2_) is manufactured worldwide as crystalline and amorphous forms for multiple applications, including tissue engineering, but our study proposes analyzing the impact of crystalline phases of TiO_2_ on Mesenchymal Stem Cells (MSCs). Several studies have already described the regenerative potential of MSCs and TiO_2_ has been used for bone regeneration. In this study, polydispersity index and sizes of TiO_2_ nanocrystals (NCs) were determined. Adipose tissue-derived Mesenchymal Stem Cells (AT-MSCs) were isolated and characterized in order to evaluate cellular viability and the internalization of nanocrystals (NCs). All of the assays were performed using the TiO_2_ NCs with 100% anatase (A), 91.6% anatase/9.4% rutile (AR), 64.6% rutile/35.4% anatase (RA), and 84.0% rutile/16% brookite (RB), submitted to several concentrations in 24-h treatments. Cellular localization of TiO_2_ NCs in the AT-MSCs was resolved by europium-doped NCs. Viability was significantly improved under the predominance of the rutile phase in NCs with localization restricted at the cytoplasm, suggesting that AR and RA NCs are not genotoxic and can be associated with most cellular activities and metabolic pathways, including glycolysis and cell division.

## 1. Introduction

Among nano biomaterials used in tissue engineering, titanium dioxide (TiO_2_) nanoparticles have been proposed as inducers of osteogenic differentiation in stem cells due to their increased adhesion to bone precursor cells, leading to accelerated bone formation [1]. Besides that, several authors have used this biomaterial for tissue repair. Bone tissue is commonly mentioned when using TiO_2_ nanoparticles, as in its association with chitosan films to improve their mechanical properties and, thus, be able to mimic bone tissue [2]. Another study also with chitosan, but in the form of a sponge, obtained better results in bone regeneration when TiO_2_ nanoparticles improved the mechanical and osteogenic properties of the material [3].

There are other applications of TiO_2_, such as its use for wound healing. Some authors synthesized these nanoparticles, marked them with curcumin, and placed them on an adhesive to attest in rats their effectiveness in healing [4]. Another group developed a gellan gum and combined it with the same TiO_2_ nanoparticles showing better results than treatment with only the gellan gum [5].

Tissue regeneration has been further improved with the use of mesenchymal stem cells (MSCs), which have also been used as an important tool for biocompatibility analyses [6]. Among all sources of MSCs, the adipose tissue-derived MSC (AT-MSCs) appears to be the most attractive one due to their similarity to bone marrow stem cells, easy isolation from cosmetic liposuctions, easy management under standard tissue culture conditions, in vivo and in vitro multi-lineage differentiation capacity, and therapeutic potential [6,7,8].

Several authors have already described the regenerative potential of stem cells that are derived from adipose tissue. One of the best-characterized aspects in this area is related to bone tissue regeneration. A research group compared the stem cells of the tooth pulp with those from the adipose tissue, and the latter one showed greater potential for osteogenic differentiation, greater expression of osteoblast marker genes, and greater deposition of minerals, with visible bone tissue appearing in a week [9]. Other researchers studied the regenerative capacity of these cells in bone defects that are caused by infections. They obtained good results in bone healing and were able to analyze the immunomodulatory effect of these cells [10]. In addition to applications for bone tissue, there are several studies showing other biomedical applications of these cells: the regeneration of organs, such as the bladder, regeneration of cardiac tissue after myocardial infarctions, and even regeneration of nervous tissue [11,12,13].

Although TiO_2_ NCs have shown good tissue biocompatibility, they have also displayed toxic potential. A study revealed pulmonary inflammation that is induced by the retention of ultrafine nanoparticles, which has led to suggestions that physical parameters, such as size, morphology, crystalline phase, and composition, might have favored nanoparticles toxicity [14]. Titanium dioxide is a polymorphic material with the crystalline phases anatase, rutile, and brookite. It has been considered a non-toxic mineral, which is traditionally used in cosmetics, food, and drug compositions [15,16]. Although these nanocrystals (NCs) have been applied in tissue engineering, there is still controversy regarding their cytotoxicity and genotoxicity [14], since it is classified by the International Agency for Research on Cancer as possibly carcinogenic to humans [IARC, 2010]. The cytokinesis-block micronucleus assay and the in vivo Drosophila wing spot test demonstrated that TiO_2_ NCs genotoxicity depends on the crystalline phases and their mixture [17], probably because it can be internalized into the cell in both cytoplasm and nucleus [18].

Because of the increasing risk of human exposure to TiO_2_ NCs, their expanding use on tissue engineering and the need to establish associations between their physicochemical properties and toxic effects on stem cells, we have investigated how TiO_2_ NCs and their nanocomposites with different phases, may affect adipose tissue-derived MSCs viability. Moreover, we also analyzed whether TiO_2_ can be internalized into the cell nucleus.

## 2. Materials and Methods

### 2.1. Synthesis and Characterization of Pure and Eu^+3^-Doped TiO_2_ NCs

TiO_2_ NCs were synthesized at room temperature in aqueous solution containing 300 mL of ultrapure water, 90 mL of nitric acid (HNO_3_, 70%, Sigma Aldrich, Saint Louis, MO, USA) and 60 mL of titanium isopropoxide (Ti(OCH(CH_3_)_2_)_4_, 97%). The components of the solution were mixed under magnetic stirring for 20 min. The pH was adjusted to 5.6 with 4 M sodium hydroxide (NaOH, 98%). The resulting solution was left standing for precipitation of TiO_2_ NCs. The precipitate was monodispersed in ultrapure water under magnetic stirring and then submitted to centrifugation at 6000 rpm for 10 min. The resulting purified precipitate was subjected to thermal treatments in environmental atmosphere, which have led to different phases compositions: 100 °C/24 h = 100% anatase (A), 500 °C/1 h = 91.6% anatase/9.4% rutile (AR), 650 °C/1 h = 64.6% rutile/35.4% anatase (RA), and 800 °C/1 h = 84.0% rutile/16% brookite (RB).

Eu-doped (Europium-doped) TiO_2_ NCs were synthesized based on the same methodology previously described for TiO_2_ NCs, but using europium chloride. The Eu^+3^ concentration of 1 wt% was determined based on the mass percentage of Ti present in TiO_2_. The Eu-doped TiO_2_ NCs were thermal annealing at 800 °C/1 h presenting the three crystalline phases with the best results regarding agglomerate stability and biocompatibility and containing Eu^+3^to enable luminescence monitoring. This doping was only applied to the RB NCs (800 °C), which presented the best stability and biocompatibility.

The X-ray diffraction (XRD) was recorded with a DRX-6000 (Shimadzu, Shenzhen, China) while using monochromatic radiation Cu-Kα1 (λ = 1.54056 Å) in order to confirm the formation of TiO_2_ NCs, as well as the crystal structure, size and average mass fraction of the phase. The size of the NCs was estimated based on the Debye–Scherrer equation (Guinier, A. X-ray diffraction in crystals, imperfect crystals, and amorphous bodies. W.H. Freeman: San Francisco, 1963) with 4.4 (anatase), 8.5 (anatase), 32 (rutile), and 44.1 nm (rutile) for thermal annealing at 100 °C/24 h, 500 °C/1 h, 650 °C/1 h, and 800 °C/1 h, respectively. The XRD patterns of the samples treated at 100 °C/24 h and 500 °C/1 h used the (101) Bragg diffraction peak of the anatase phase located at approximately 2θ = 25.4°, and samples that were treated at 650 °C/1 h and 800 °C/1 h used the peak (110) of the rutile phase located at approximately 2θ = 27.8°. The percentage of anatase, brookite, and rutile phases were calculated based on reports elsewhere [19,20]. The Raman spectra were recorded with a LabRAM HR Evolution Spectrometer (Horiba) using a 633 nm laser. All of the characterizations were performed at room temperature.

### 2.2. Dispersion of Nanocomposites and Eu-Doped TiO_2_ NCs

Stock solutions of 2 mg/mL (pH 4) of samples A and AR NCs were prepared in deionized (DI) water. After this procedure, an ultrasonic disintegrator (Q-Sonica 700 W, Newtown, CT, USA) was used to disperse the NCs at 50% amplitude for 15 min. For RA and RB NCs (with or without Eu^+3^), stock solutions of 10 mg/mL (pH4) were prepared in DI water and dispersion was carried out for 10 min. at 30% amplitude.

Dispersed stock solutions were diluted in DMEM culture medium (Dulbecco’s modified Eagle’s medium—Low glucose) (Cultilab, São Paulo, Brazil) supplemented with 10% (*v*/*v*) fetal bovine serum (FBS) (Cultilab, São Paulo, Brazil) at concentrations of 5, 50, 100, and 250 µg/mL for their use in cell culture. Bovine serum albumin (BSA) was used as a stabilizing agent due to the increase of NC aggregation in the culture medium, according to protocol optimized elsewhere [21].

### 2.3. Transmission Electron Microscopy (TEM)

The NC samples were diluted in DI water, sonicated and allowed to stand for 3 h. The supernatant was withdrawn, which was considered to be the most stable, monodispersed colloidal NCs. The resulting suspensions were dripped onto lacey carbon films on 300 mesh copper grids.

A Tecnai G2 Spirit Twin (FEI Company, Hillsboro, OR, USA) transmission electron microscope with a lanthanum hexaboride emitter was used in order to observe the morphology, sizes, and crystalline phases of NCs and their agglomerates. Multi-beam, bright-field, and dark-field images were taken at 120 kV. The images of electron diffraction patterns were then analyzed and indexed in order to determine the composition of crystalline phases of NCs by MacTempas software (Total Resolution, Berkeley, CA, USA).

### 2.4. Dynamic Lighting Scattering (DLS) Analysis

DLS measurements were performed using the Zetasizer Nano ZS (Malvern Instruments, Malvern, UK) operating in retraction mode at an angle of 173° and automatically determined by the Zetasizer software. Dispersed stock solutions of the four NC samples were sonicated in DI water and run for the same length of time. The agglomerates and the polydispersity index (PDI) of the NCs were measured in complete culture medium (DMEM, 10% FBS, and BSA). NCs were prepared in the same way for incubation with AT-MSCs.

### 2.5. Isolation and Characterization of Human AT-MSCs

AT-MSCs were acquired through liposuction of one patient who signed an informed consent form according to the Project No. 1.776.680 approved by the Human Ethics Committee at UFU.

The material was allowed to stand for 1 h for decantation, separating the blood at the bottom and oil at the top, and then 15 mL of adipose cells were transferred into a new tube. Adipose tissue was washed three times with FBS (15 mL), followed by centrifugation at 1500 rpm for 5 min. Adipose tissue fragments were collected, seeded in the corners of T25 flasks, and covered with about 2.5 mL of FBS. After 24 h, DMEM supplemented with 20% FBS was added. The cells were maintained at 37 °C in a humidified incubator with 5% CO_2_. The medium was changed every two days until cells reached 80–90% confluence.

The markers used to confirm the AT-MSCs were: APC-CD11b, FITC-CD45, PerCP-CD34, and PE-CD73 (Beckman Coulter, Brea, CA, USA), according to the International Society for Cellular Therapy [22]. The samples were analyzed on the LSRFortessa flow cytometer (BD Biosciences, San Jose, CA, USA) using BD FACS Diva and FlowJo software (7.6.5, BD Biosciences, San Jose, CA, USA).

### 2.6. Cell Viability Assay using Hoechst 33342 and PI^+^

AT-MSCs at 80–90% confluence were trypsinized (Trypsin-EDTA, Gibco, Big Cabin, OK, USA), seeded into 96-well microplates (1 × 10^4^ cells per well), and then incubated in DMEM with 10% FBS for 24 h at 37 °C and 5% CO_2_. After 24 h, the culture medium was replaced by treatment with NCs solutions (A, AR, RA and RB) at different concentrations of 5, 50, 100, and 250 μg/mL for 24 h. After treatments, cells were washed once with phosphate buffer solution.

The simultaneous staining with 50 µg/mL of Propidium iodide (PI^+^) (Sigma–Aldrich, Saint Louis, MO, USA) and Hoechst^®^ 33342 16 μM (Thermo Scientific, Waltham, MA, USA) was used in each well for analysis of cell viability, which was incubated for 10 min. Images were acquired using the Operetta High Content System (Perkin Elmer, Waltham, MA, USA). To evaluate the viability, the cell number was determined by nuclei counting using the Hoechst^®^ 33342 fluorescence channel. Viability was calculated as the ratio of the number of nuclei stained with PI^+^ (dead cells) and the number of nuclei stained with Hoechst 33342 (dead and living cells). This automated cell counting was performed by Harmony software version 3.5.2 (Perkin Elmer) using an algorithm provided by the software building blocks [23].

Morphological alterations regarding cell area, symmetry, width, length, and width versus length parameters were calculated using the Harmony Software version 3.5.2 (Perkin Elmer).

### 2.7. Localization Assay of Eu-Doped TiO_2_ NCs

Trypsinized AT-MSCs were seeded on 13-mm circular coverslips into 24-well microplates (0.5 × 104 cells per well) and incubated in DMEM with 10% FBS for 24 h at 37 °C and 5% CO_2_. The cells were treated with Eu-doped TiO_2_ NCs at different concentrations (5, 50, 100 and 250 μg/mL) and incubated for 24 h. The coverslips were fixed with 4% paraformaldehyde and mounted on slides with DAPI dye (ProLong™ Gold AntifadeMountant with DAPI, Thermo Fisher, Waltham, MA, USA). Confocal A1+ (Nikon, Tokyo, Japan) microscope was used in order to visualize the NCs and ImageJ program was used to quantify the NCs according to their fluorescence intensity (emission and excitation spectra of Eu doped TiO_2_ NCs are 390–610 nm).

### 2.8. Statistical Analysis

The statistical analysis for mean comparisons among viability and cytotoxicity assays were performed by the GraphPad Prism 7.4 (2018) software (7.4, GraphPad Software, San Diego, CA, USA) using the two-way ANOVA method. All of the experiments were performed in triplicate.

## 3. Results

### 3.1. Characterization of Nanocrystals

The structural and optical properties of pure and Eu^-^doped TiO_2_ NCs were investigated while using the X-ray diffraction (XRD) and photos without and with ultraviolet light, respectively. Figure 1 shows the XRD patterns of (a) TiO_2_ NCs (b) and Eu-doped TiO_2_ NCs, as well as (c) photos of samples without and with ultraviolet light.

For the X-ray diffraction (XRD) patterns (Figure 1a), observed that all samples are formed by TiO_2_ nanocrystals. The thermal annealing at 100 °C/24 h and 500 °C/1 h shows Bragg diffraction peaks characteristic of anatase phase (JCPDS, No. 21-1272). However, for the sample at 650 °C/1 h and 800 °C/1 h, observed Bragg diffraction peaks of rutile and anatase phases (rutile: JCPDS, No. 76-1940), and rutile and brookite phases (brookite: JCPDS, No. 76-1940), respectively. The percentage of crystalline phase present in the samples with thermal annealing at 500 °C/1 h (A), 650 °C/1 h, and 800 °C/1 h were 91.6% anatase/9.4% rutile (AR), 64.6% rutile/35.4% anatase (RA), and 84.0% rutile/16% brookite (RB), respectively. The size and percentage of crystalline phases for the Eu-doped TiO_2_NCs were similar to those of TiO_2_ NCs, as observed in the XRD pattern (Figure 1b).

The photos of pure and Eu-doped TiO_2_ NCs without and with ultraviolet light are shown in Figure 1c. In the absence of ultraviolet light, the samples are white powders. However, in the presence of ultraviolet showed red luminescence, characteristic of Eu^+3^ ions when compared to the images in the absence of ultraviolet light and the non-doped counterparts.

### 3.2. Morphological Properties

Ultrastructural imaging using transmission electron microscopy (TEM) was performed in order to investigate the size, crystalline phases and agglomeration of these nanocrystals (Figure 2). It is important to highlight that the concentration of nanocrystals solutions used was the same in the DLS and biological assays, which has led to agglomerates of NCs. TEM is the most adequate technique to observe the morphology, size, and crystalline structure of samples of nanometric particles. It is important to highlight that, during sample preparation, NCs solutions were dripped onto carbon films, forming agglomerates of NCs after drying. The four samples (A, AR, RA, and RB) displayed several morphologies and crystalline phases.

Sample A had regions with anatase NCs of homogeneous distribution and size of approximately 10–20 nm (Figure 2A(a)), with regions varying sizes, ranging from 10 to 200 nm (Figure 2A(b)). The electron diffraction pattern of sample A displayed rings that corresponded to the anatase structure with low definition, formed by low-intensity points (Figure 2A(c)). This fact indicates that the material had low crystallinity, in agreement with the results of the XRD pattern that also showed low-intensity diffraction peaks and no diffraction rings of the rutile structure were observed (Figure 1A).

The AR sample had a homogeneous size distribution approximately between 10 and 20 nm (Figure 2B(a)) and particles lacked a well-defined morphology (Figure 2B(b)). The electron diffraction pattern of sample AR (Figure 2B(c)) showed typical rings of anatase structure. As the percentage of the rutile phase was relatively small, based on XRD results (Figure 1A), this justifies the lack of observation of diffraction rings of the rutile structure.

In the TEM image of the sample, RA showed the coexistence of two well-defined NC species that can be differentiated by both the bimodal size distribution and presence of two crystal structures in the electron diffraction pattern. Particles of smaller size, between 10–40 nm, corresponded to the anatase phase, and larger particles, between 100–200 nm, corresponded to the rutile phase (Figure 2C(a)). The particles of A were in greater number than the R, but particles of R occupied a much greater volumetric fraction, although in smaller concentration (Figure 2C(b)). In the electron diffraction pattern (Figure 2C(c)), diffraction rings corresponded to the structures of A (body-centered tetragonal) and R (simple tetragonal) were observed. The rings corresponding to A were formed by a greater number of points than those corresponding to R, which were formed in isolated points.

In the TEM image of the sample, RB showed a size distribution of NCs of the order of 50–100 nm (Figure 2D(a)), with a very small number of particles of the order of 5–10 nm (Figure 2D(b)). The electron diffraction pattern (Figure 2D(c)) showed points that corresponded to the R and brookite (B) structures. The plane (200) of the orthorhombic phase of B of 4.59 Å of interplanar distance was identified and most of the diffraction points corresponding to the R phase.

### 3.3. Dynamic Lighting Scattering (DLS) Analysis

DLS was used to determine the mean particle size or particle size distribution (PSD) of NCs suspended in different media. DLS was also used to measure the agglomerate sizes of TiO_2_ NCs. NCs were sonicated in culture complete medium (DMEM + BSA + FBS) and deionized water (DI) and data were collected, including the NCs distribution, according to their intensity, area, volume, quantity, and polydispersity index (PDI).

It is important to note that TiO_2_ NCs were not functionalized during synthesis, in order to investigate the real biological effects of TiO_2_ nanocrystals. Thus, the aggregation of nanocrystals occurs due to van der Waals interactions. Thus, the size observed in the DLS measurements will be quite different from those that were obtained in the transmission images.

PDI measures the homogeneity of particle sizes and indicates the size distribution of NCs ranging from 0 to 1. Values close to zero indicate a homogeneous dispersion while those greater than 0.3 indicate high heterogeneity [24]. We observed that the A and RB NCs PDI (0.27 and 0.20 respectively) showed values below 0.3 indicating high size homogeneity in the complete medium (Table 1). The samples of NCs RB showed the lowest PDI in complete medium, meaning that they had high homogeneity (Figure 3A).

TiO_2_ NCs showed a similar size agglomerate, independent of the type of NCs (A, AR, RA, and RB), when sonicated in complete medium (Table 1). A, AR, RA, and RB NCs showed statistical difference when the NCs were suspended in DI as compared with NCs that were suspended in complete medium (Figure 3B), meaning that NCs suspended in complete medium had low heterogeneity.

### 3.4. Isolation and Characterization of Human AT-MSCs

After isolation and initiation of cell culture, the cells showed a fibroblastoid morphology in monolayer culture and adherence to the plastic surface, which correspond to the first requirements for the characterization of MSCs (Figure 4). Immunophenotyping was performed by the evaluation of CD45 (0.3%), CD34 (0.6%), and CD11b (10.8%) as negative, and CD73 (96.6%) and CD90 (96.3%), as positive surface markers.

### 3.5. Cell viability Assay Using an Automated Imaging Analysis System

The viability of AT-MSCs incubated with TiO_2_ NCs was evaluated 24 h after treatment. Imaging analysis for quantification of NCs cytotoxicity was performed by an automated counting and detection of PI^+^ (propidium iodide) nuclei. The images were acquired using a high content system. After nuclei and cytoplasmatic segmentation, it was possible to detect and quantify PI^+^ nuclei in the AT-MSCs by applying specifics algorithms from Harmony software.

Comparing the viability of the negative control (without NCs) with the cells that were treated with different concentrations of NCs, it was observed that there was a statistically significant reduction in cell viability at the concentrations of 5, 50, and 100 μg/mL in the cells treated with sample A NCs (Figure 5A). In the cells that were treated with ARNCs, the concentrations of 100 and 250 μg/mL showed a statistically significant reduction in cell viability when compared to the control and concentration of 5 μg/mL (Figure 5B). Cells treated with RA NCs did not show statistical differences among different conditions (Figure 5C). Cells that were treated with RB NCs showed a statistically significant reduction in cell viability at the concentration of 250 μg/mL, when compared to the control and concentrations of 5 and 50 μg/mL (Figure 5D).

Stained cells with Hoechst 33342, a blue-fluorescence dye (excitation/ emission maxima ~350/461 nm) and Propidium iodide (PI^+^), a red-fluorescence dye (excitation/emission maxima ~535/617 nm) were collected by the Operetta High Content System (Figure 6) and confirmed the observations that were made by the cell counting assay. The majority of cells appeared in blue because the cell viability was higher than 80%.

In addition, we evaluated the presence of morphological alterations regarding cell area, symmetry, width, length, and width versus length parameters (Figure 7A). Cells that were treated with the concentrations of 100 and 250 μg/mL showed a smaller cell area (Figure 7A(a)) and width (Figure 7A(c)). Cells at the concentration of 250 μg/mL of RB NCs displayed greater symmetry than the rest (Figure 7A(b)). Cells showed greater length at all the tested concentrations when compared to the control. The higher the concentration of NCs, the shorter the length (Figure 7A(d)). Cells also showed a smaller width versus length parameter at the highest concentrations of NCs (100 and 250 μg/mL). At the concentration of 5 µg/mL of RB NC, the width versus length parameter also decreased when compared to cells at the concentrations of 100 e 250 µg/mL (Figure 7A(e)).The images made by the Operetta High Content System showed morphological changes in AT-MSCs after 24-h treatment with RB NCs (Figure 7B).

### 3.6. Localization Assay of Eu-Doped TiO_2_ NCs

Eu-doped TiO_2_ NCs with the sample of RB NCs were incubated with AT-MSC at different concentrations (5, 50, 100, and 250μg/mL) for 24 h to evaluate their capacity of internalization into these cells. We chose RB NCs due to their stability according to the DLS assay, their low cytotoxicity (allowing high cell viability), and their composition that lacks anatase, the crystalline phase with greater cytotoxicity, and genotoxicity [17].

After AT-MSCs treatment with RB NCs for 24 h, NCs were located in the cytoplasm of cells, without entering the nucleus, not only suggesting lack of genotoxic activity, but also its possible association with most cellular activities and metabolic pathways, including glycolysis and cell division.

An internalization pattern was observed in the cytoplasm of AT-MSCs. The amount of internalized NCs did not show statistical differences among different conditions (Figure 8A,B).

## 4. Discussion

Titanium dioxide (TiO_2_) is manufactured worldwide as crystalline and amorphous forms for multiple applications, including tissue engineering, but no studies have analyzed the impact of TiO_2_ crystalline phases on MSCs. In this study, we evaluated the effect of the crystalline phase TiO_2_ Nanocrystals in cytotoxicity, morphological changes, and cell internalization into human AT-MSCs, and demonstrated that, although TiO_2_ NCs are highly biocompatible, viability can be significantly improved under the predominance of rutile phase in TiO_2_ NCs.

Tissue engineering with applications of titanium dioxide (TiO_2_) nanoparticles have been proposed as inducers of osteogenic differentiation of stem cells aiming towards bone formation and tissue regeneration [1,2,3,4,5], but the effect of crystalline phases of these nanoparticles on AT-MSCs have not been investigated. Concerns about their toxicity have been raised due to their toxicity potential in lungs conferred by ultrafine nanoparticles retention, probably due to physical parameters, such as size, morphology, crystalline phase, and composition [14]. This was further corroborated by in vivo analysis of the Drosophila wing spot model, which has shown that TiO_2_ NCs genotoxicity depends on crystalline phases and their mixture [17], probably due to their internalization into the cell cytoplasm and nucleus [25,26].

To the best of our knowledge, this is the first investigation that shows the effect of crystalline phases of TiO_2_ NCs on AT-MSCs. We have first characterized the nanocrystals based on their size, crystalline phase, aggregation, and stability in order to demonstrate their impact. This may have further implications in the presence of serum, which may modulate the efficiency of internalization, consequently leading to adverse effects [27]. We have shown that the mixed phase rutile/brookite in TiO_2_ NCs was remarkably stable, since its dispersion in complete medium revealed greater stability in comparison to the others, and this feature is extremely important for biological applications.

Other studies using BSA in NCs preparation have also shown greater size stability of agglomerates [28,29]. Therefore, decreased agglomeration and sedimentation of TiO_2_ NCs mediated by proteins may be a strategy to avoid toxicity [28]. However, we have shown that increased toxicity may be due to the presence of the anatase phase in higher proportion, which led to a greater instability of preparations, even though it continued to be biocompatible.

In some studies, the anatase phase promoted greater cellular damage with severe morphological alterations, which is in agreement with other studies that also showed greater cytotoxic and genotoxic effects of anatase when compared to rutile phase [17]. This is also supported elsewhere, in which the anatase phase was 100 times more toxic than an equivalent sample of the rutile phase [30]. Therefore, it is important to consider that the crystallinity of the rutile phase may be a key factor for the lower toxicity [31,32], which we have further confirmed by showing that RB NCs only presented reduced cell viability at the highest concentration (250 µg/mL).

Regarding cell internalization, endocytosis is one of the pathways by which nanocrystals can be internalized. It has been reported elsewhere that 50 nm NPs are rapidly internalized by this process when compared to smaller (<14 nm) or larger NPs (up to 500 nm) [33], which is corroborated by our results with RB NCs, whose sizes were between 50 to 100nm. It is interesting to observe that all nanocrystals with the anatase phase component had very small sizes, which may also be associated with increased toxicity and internalization. Additionally, the uptake of nanomaterials is known to significantly affect the shape, size, and viability of cells, which is due to the type and concentration of nanomaterials [34]. Our work demonstrated significant differences in cellular morphology (symmetry) at 250 μg/mL of RB NCs, corroborating the notion that the toxicity of the safest TiO_2_ nanocrystal form may also occur under high concentration.

Finally, the next key matter post-NCs uptake is their intracellular trafficking, which will determine their destination within cellular compartments as well as their cytotoxicity. It has been shown elsewhere that TiO_2_ NCs induce significant DNA damage [26] due to their internalization into the cells nuclei. The authors used the anatase phase NCs, and this may have probably been the reason for the cytotoxicity. On the contrary, we have shown that RB NCs with rutile phase predominance presented restricted localization to the cytoplasm. Studies using europium doped in the different nanoparticles showed that no luminescent nanoparticles were observed within the nuclei of the cells [35,36], corroborating with our results and suggesting that these TiO_2_ NCs are not genotoxic and may be associated with most cellular activities and metabolic pathways, including glycolysis and cell division.

## 5. Conclusions

This is the first report about the impact of the crystalline phases of TiO_2_ nanocrystals on human adipose tissue-derived mesenchymal stem cells. All of the pure and mixed crystalline phase of TiO_2_ NCs showed great cell biocompatibility. However, viability was better under the predominance of the rutile phase in the concentrations of 5, 50 and 100 μg/mL. Whereas, cell morphological changes were significantly affected under low concentrations of TiO_2_ NCs with a predominance of anatase phase, changes were detected for the rutile phase only in the highest concentration. Additionally, the restricted internalization of Eu-doped TiO_2_ NCs with mixed phase rutile/brookite only into the cytoplasm further corroborates their low cytotoxic and genotoxic effects, ideal features of nanocrystals for tissue engineering applications.

## Figures and Tables

**Figure 1 materials-13-04071-f001:**
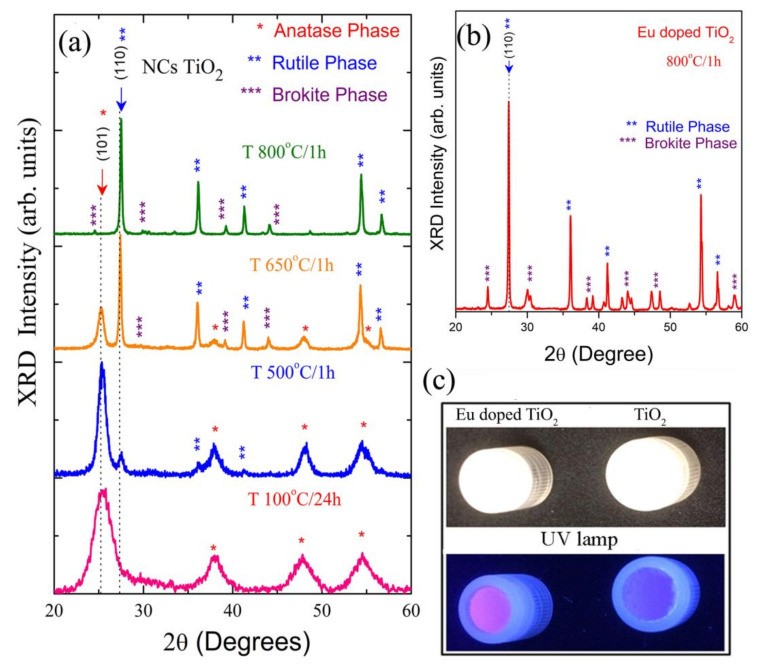
(**a**) X-ray diffraction (XRD) patterns of Titanium dioxide (TiO_2_) nanocrystals (NCs), (**b**) XRD pattern of Eu-doped at 1 wt% concentration Rutile/Brokite NCs. Eu-doped TiO_2_ NCs were annealing thermal at 800 °C/1 h presented the three best crystalline phases, and (**c**) photos of samples without and with ultraviolet light showing that the Eu-doped TiO_2_ enabled luminescence monitoring. In the absence of ultraviolet light, the samples are white powders. However, in the presence of ultraviolet showed red luminescence, characteristic of Eu^+3^ ions.

**Figure 2 materials-13-04071-f002:**
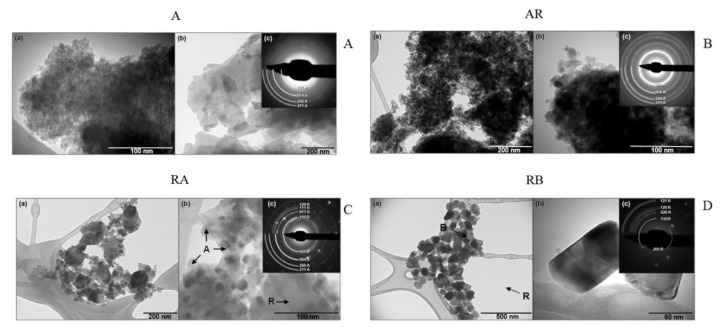
Transmission electron microscopy (TEM) analysis of the size, crystalline phases, and agglomeration of NCs. (**A**) Sample A: (**a**) region with anatase NCs of homogeneous size distribution, (**b**) region with a heterogeneous size distribution, (**c**) electron diffraction pattern of anatase poorly defined. (**B**) Sample AR: (**a**) NCs of anatase of homogeneous size, (**b**) area with NCs ata higher magnification, (**c**) electron diffraction pattern typical of anatase structure. (**C**) Sample RA: (**a**) small agglomerate containing anatase and rutile NCs, (**b**) image identifying anatase (A) (smaller particles) and rutile (R) (larger particles) at a higher magnification, (**c**) electron diffraction pattern with rings corresponding to anatase (A) and rutile (R) structures. (**D**) Sample RB: (**a**) small agglomerate of NCs of homogeneous size, (**b**) image of NCs at a higher magnification, (**c**) electron diffraction pattern containing rings of rutile (R), and brookite (B).

**Figure 3 materials-13-04071-f003:**
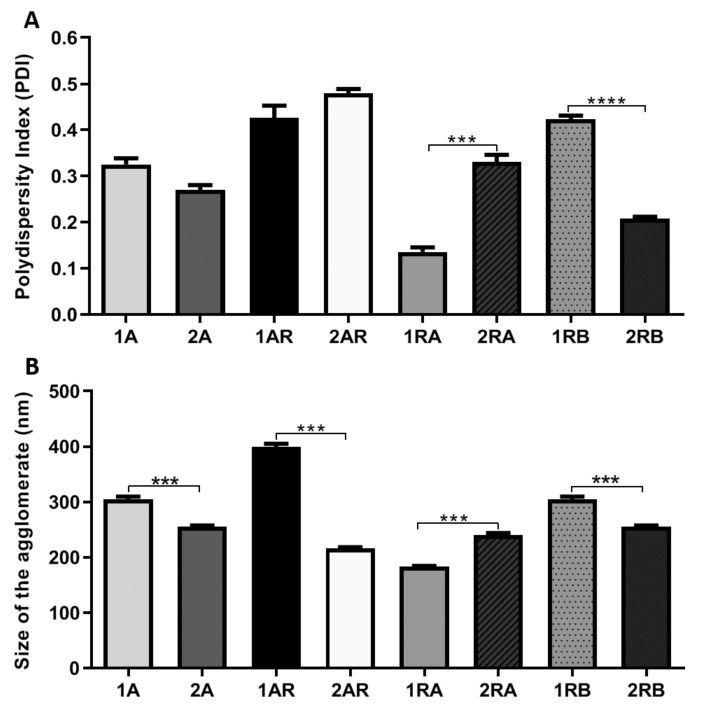
(**A**) Polydispersity index (PDI) values of the four types of NCs (A, AR, RA, and RB) in DI (1A, 1AR, 1RA, and 1RB), and complete medium (2A, 2AR, 2RA and 2RB). (**B**) Size of the Agglomerates according to DLS analysis of the four types of TiO_2_ NCs when sonicated in DI water (1A, 1AR, 1RA, and 1RB) and complete medium (2A, 2AR, 2RA, and 2RB). Statistical differences were calculated using the two-way ANOVA method, where *** *p* < 0.0005 and **** *p* < 0.00005. All of the experiments were performed in triplicate.

**Figure 4 materials-13-04071-f004:**
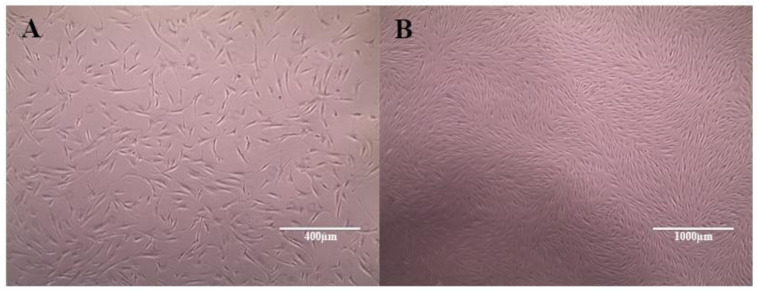
Adipose tissue-derived Mesenchymal Stem Cell (AT-MSC) morphology in the second passage: adherent and fibroblastoid morphology. Photograph obtained by electron microscopy at 10 × (**A**) and 4 × (**B**) magnifications, respectively.

**Figure 5 materials-13-04071-f005:**
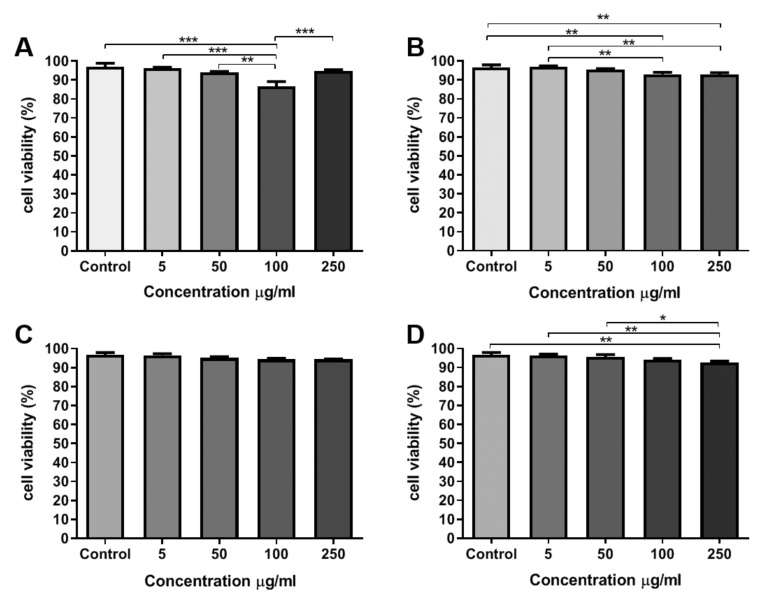
AT-MSCs viability after treatment with TiO_2_ NCs in different concentrations. (**A**) Cells treated with sample A. (**B**) Cells treated with sample AR. (**C**) Cells treated with sample RA. (**D**) Cells treated with sample RB. Statistical differences were calculated using the two-way ANOVA method, where * *p* < 0.05, ** *p* < 0.005 and *** *p* < 0.0005. All of the experiments were performed in triplicate.

**Figure 6 materials-13-04071-f006:**
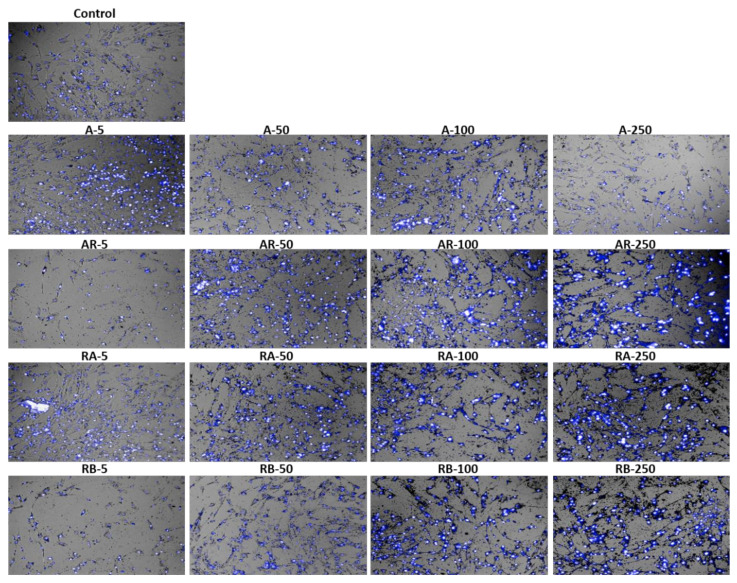
Microscopic images of AT-MSCs after treatment with TiO_2_ nanocrystals. Cells treated with samples of A NCs, AR NCs, RA NCs, and RB NCs. PI (dead cells) and Hoechst 33342 (dead and live cells) double-staining. Control: cells without treatment. Photograph obtained by the high content equipament (fluorescence microscopy) at 20× magnification.

**Figure 7 materials-13-04071-f007:**
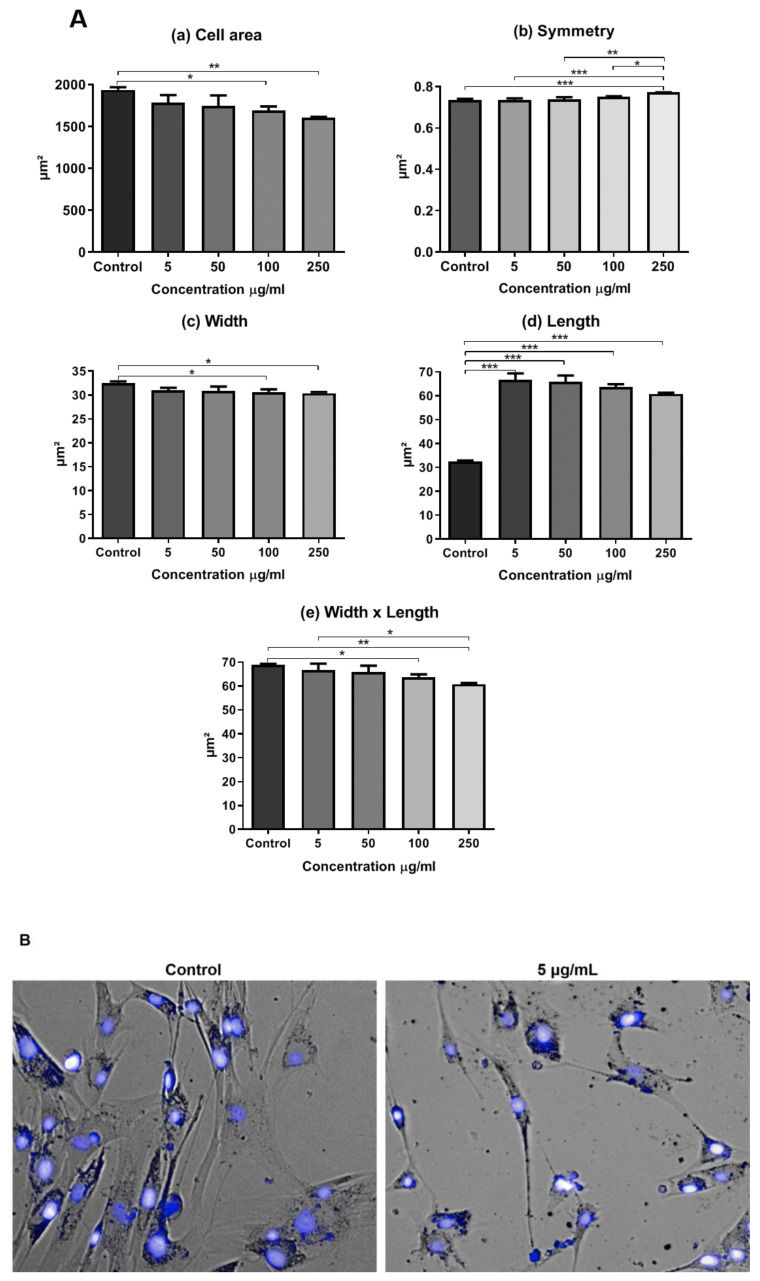
Morphology of AT-MSCs. (**A**) Morphological analysis when treated with RB NCs. (**a**) Cell area. (**b**) Symmetry. (**c**) Width. (**d**) Length. (**e**) Width versus length. (**B**) Images of AT-MSCs taken by the High Operetta Content System. Untreated (control) cells and treated cells with RB NCs at the concentration of 5 μg/mL. Photograph obtained by electron microscopy at 20 × magnification. Statistical differences were calculated using the two-way ANOVA method, where * *p* < 0.05, ** *p* < 0.005 and *** *p* < 0.0005. All experiments were performed in triplicate.

**Figure 8 materials-13-04071-f008:**
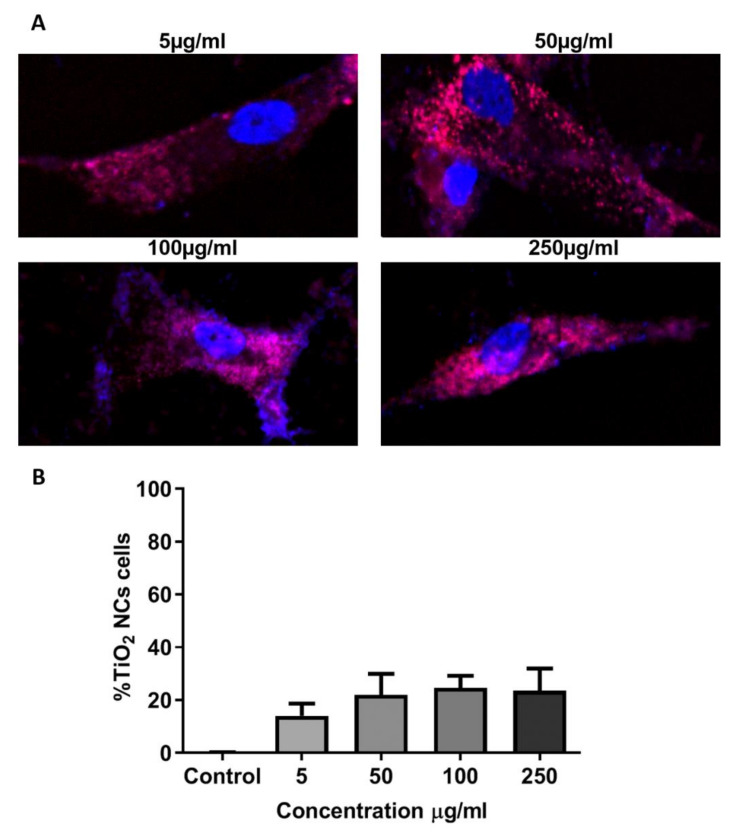
(**A**) Fluorescent imaging of AT-MSCs after 24 h treatment concentrations with Eu-doped RB NCs in 5, 50, 100, and 250 μg (40×). (**B**) Quantification of sample Eu-doped RB NCs in AT-MSCs after 24 h treatment (ImageJ program). All experiments were performed in triplicate.

**Table 1 materials-13-04071-t001:** Average values of the polydispersity index (PDI) values of the four types of NCs in DI and complete medium and the size of the agglomerate of TiO_2_ NCs analyzed by Dynamic Lighting Scattering (DLS) after sonication in deionized (DI) water and complete medium.

TiO_2_ NCs Sample/Phase	PDI DI	PDI Complete Medium	Size DI	Size Complete Medium
A	0.41	0.27	270 nm	232 nm
AR	0.42	0.47	400 nm	217 nm
RA	0.13	0.33	184 nm	240 nm
RB	0.42	0.20	305 nm	256 nm

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
