# Peer review of "Characterization of Crystalline Phase of TiO2 Nanocrystals, Cytotoxicity and Cell Internalization Analysis on Human Adipose Tissue-Derived Mesenchymal Stem Cells"

_materials, 2020, doi:10.3390/ma13184071_

Round 1

Reviewer 1 Report

The authors investigated the cytotoxicity of four different TiO2 nanocrystals on human adipose tissue-derived mesenchymal stem 4 cells, and evaluated the effect of crystalline phase on the performance. The result might be of interest to the researchers in the field of tissue engineering. I recommend its publication in Materials after the following comments are addressed:

  1. The language and format of the manuscript should be carefully checked, as I found a number of mistakes and typos. For example, “but no studies have analyzed the 31 impact crystalline phases of TiO2 on Mesenchymal Stem Cells”, “3.1 Structuralandopticalproperties”, “Eu+3doped”, etc.

  1. In figure 1, the detailed info should be given in the caption, such as the Eu fraction/concentration, the sample form of the nanocrystals, and the container for the photos. Meanwhile, the photos should be taken with non-emissive container to prevent the interference.

  1. In Figure 2, the scale bar should be made clear with recognizable size number, and average size should be calculated from the TEM images. Then the values can be compared with the DLS results in Table 1.

  1. In table 1, the size cannot be precise to 0.01nm, the numbers after the decimal point should be removed. In addition, standard deviation should be provided.

  1. The cell images in Figure 6 and Figure 8 are too small to read. What’s the fluorescence spectra of Eu doped TiO2 crystals? Did the authors obtained the cell images based on the FL microscopy. Would the Eu doping affect the internalization?

Author Response

1- The language and format of the manuscript should be carefully checked, as I found a number of mistakes and typos. For example, “but no studies have analyzed the 31 impact crystalline phases of TiO2 on Mesenchymal Stem Cells”,“3.1 Structuralandopticalproperties”, “Eu+3doped”, etc.

Response: The mistakes and typos were revised and corrected.

2- In figure 1, the detailed info should be given in the caption, such as the Eu fraction/concentration, the sample form of the nanocrystals, and the container for the photos. Meanwhile, the photos should be taken with non-emissive container to prevent the interference.

Response: Thanks for the comment. We added the sentence below in the Materials and Methods section and in the caption:

“Eu-doped TiO2 NCs were synthesized based on the same methodology as TiO2 NCs, previously, but with europium chloride. The Eu concentration of 1 wt% was determined based on the mass percentage of Ti present in TiO2. The Eu-doped TiO2 NCs were thermal annealing at 800oC/1h presented the three crystalline phases with best results agglomerate stability and biocompatibility and containing Eu to enable luminescence monitoring.”

The container used for the photos was the plastic cover.

3- In Figure 2, the scale bar should be made clear with recognizable size number, and average size should be calculated from the TEM images. Then the values can be compared with the DLS results in Table 1.

 Response:  The figure 2 scale bar was corrected. Thanks for the comment. However the values of the nanocrystals were determined as shown in the following paragraphs highlighted.

“Sample A had regions with anatase NCs of homogeneous distribution and size of approximately 10-20 nm (Figure 2A (a)), with regions varying sizes from 10 to 200 nm (Figure 2A (b)). The electron diffraction pattern of sample A displayed rings corresponding to the anatase structure with low definition, formed by low-intensity points (Figure 2A (c)). This fact indicates that the material had low crystallinity, in agreement with the results of the XRD pattern that also showed low-intensity diffraction peaks and no diffraction rings of the rutile structure were observed (Figure 1A).

The AR sample had a homogeneous size distribution approximately between 10 and 20 nm (Figure 2B (a)) and particles lacked a well-defined morphology (Figure 2B (b)). The electron diffraction pattern of sample AR (Figure 2B (c)) showed typical rings of anatase structure. As the percentage of the rutile phase was relatively small, based on XRD results (Figure 1A), this justifies the lack of observation of diffraction rings of the rutile structure.

In the TEM image of the sample, RA showed the coexistence of two well-defined NC species that can be differentiated by both the bimodal size distribution and the presence of two crystal structures in the electron diffraction pattern. Particles of smaller size, between 10-40 nm, corresponded to the anatase phase, and larger particles, between 100-200 nm, corresponded to the rutile phase (Figure 2C (a)). The particles of A were in greater number than the R, but particles of R occupied a much greater volumetric fraction, although in smaller concentration (Figure 2C (b)). In the electron diffraction pattern (Figure 2C (c)), diffraction rings corresponded to the structures of A (body-centered tetragonal) and R (simple tetragonal) were observed. The rings corresponding to A were formed by a greater number of points than those corresponding to R, which were formed in isolated points.

In the TEM image of the sample, RB showed a size distribution of NCs of the order of 50-100 nm (Figure 2D (a)) with a very small number of particles of the order of 5-10 nm (Fig. 2D(b)). The electron diffraction pattern (Figure 2D (c)) showed points corresponding to the R and brookite (B) structures. The plane (200) of the orthorhombic phase of B of 4.59 Å of interplanar distance was identified and most of the diffraction points corresponding to the R phase.

4- In table 1, the size cannot be precise to 0.01nm, the numbers after the decimal point should be removed. In addition, standard deviation should be provided.

Response: The numbers after the decimal point were removed. We have added in the figure 3 another graphic (figure 3A e 3B) with the statistical analyses showing the standard deviation for polydispersity index (PDI) and the text was improved: “PDI measures the homogeneity of particle sizes and indicates the size distribution of NCs ranging from 0 to 1. Values close to zero indicate a homogeneous dispersion while those greater than 0.3 indicate high heterogeneity [15]. We observed that the A and RB NCs PDI (0.27 and 0.20 respectively) showed values below 0.3 indicating high size homogeneity in the complete medium (table 1). The samples of NCs RB showed the lowest PDI in complete medium, meaning that they had high homogeneity (figure 3A).

TiO2 NCs showed a similar size agglomerate, independently of the type of NCs (A, AR, RA and RB), when sonicated in complete medium (table 1). A, AR, RA and RB NCs showed statistical difference when the NCs had been suspended in DI as compared with NCs suspended in complete medium (Figure 3B), meaning that NCs suspended in complete medium had low heterogeneity.”

5- The cell images in Figure 6 and Figure 8 are too small to read. What’s the fluorescence spectra of Eu-doped TiO2 crystals? Did the authors obtained the cell images based on the FL microscopy. Would the Eu doping affect the internalization?

Response: We increased the size of the figures 6 and 8 and we kept only in the figure 8 images with PI and Hoeschst staining. Hoechst 33342 is a kind of blue-fluorescence dye (excitation/ emission maxima ~350/461 nm).When bound to DNA, it stains the condensed chromatin in apoptotic cells more brightly than the chromatin in normal cells. Propidium iodide (PI) is a red-fluorescence dye (excitation/emission maxima ~535/617nm when bound to DNA), only permeable to dead cells. The majority of cells were in blue because the cell viability was higher than 80%.In figures 6 and 8, Eu doping was not used. Only in the experiments for internalization we used Eu doped TiO2 NCs.

We changed the text of the results and methodologies to clarify these information and the numbers of the figures. Figure 7: Morphology of AT-MSCs and figure 8A: Fluorescent imaging of AT-MSCs after 24h treatment concentrations with Eu-doped RB NCs in 5, 50, 100 and 250 μg (40x).  8B: Quantification of sample Eu-doped RB NCs in AT-MSCs after 24h treatment (ImageJ program).

The typical emission and excitation spectra of Eu doped TiO2 NCs are 390-610nm. The figure below shows FL spectra of TiO2 pure and doped with Eu3+. The TiO2 NCs emit in blue and with the doping of Eu3+ observed a characteristic luminescence predominantly in red.

                       Figure. FL spectra of TiO2 pure and doped with Eu3+.

The images for internalization analysis had been performed by FL microscopy. This information is in the Material and Methods section: “Confocal A1+ (Nikon, Japan) microscope was used to visualize the NCs and ImageJ program was used to quantify the NCs according to their fluorescence intensity”.

Our results are according studies where Eu doping don´t affect the internalization. Doat et al (2003) studied mineral nanoparticles of apatitic tricalcium phosphate doped with europium, they incubated human pancreatic epithelial cells with these particles and their internalization was observed by laser scanning confocal microscopy, and confirmed by Transmission electron microscopy and electron microdiffraction analysis that the particles were internalized retaining their original apatitic structure. They showed that no luminescent nanoparticles were observed within the nuclei of the cells (doi:10.1016/S0142-9612(03)00169-8). Wong et al (2008) showed results of the functionalized europium nanorods (FENR) in human lung carcinoma A549 and Hela cells and the FENR was not observed inside the nuclei, only inside the cell cytoplasm (https://doi.org/10.1021/ic8000416).

The text below was added in Discussion section:

“Studies using europium doped different nanoparticles showed that no luminescent nanoparticles were observed within the nuclei of the cells (doi:10.1016/S0142-9612(03)00169-8 and https://doi.org/10.1021/ic8000416), corroboranting with our results and suggesting that these TiO2 NCs are not genotoxic and may be associated with most cellular activities and metabolic pathways, including glycolysis and cell division”.

Reviewer 2 Report

The paper entitled “Caracterization of Crystalline Phase of TiO2 Nanocrystals, cytotoxicity cell internalization analyse  on human adipose tissue-derived mesenchymal stem cells” provides an interesting study for the researchers working on nanomaterials with biomedical applications. However, several points must be addressed before being considered for publication.

  1. The introduction section is too short and must be improved. For example, examples of biomedical aplications of ADSC should be provided such as bladder (https://doi.org/10.1093/rb/rbz049; https://doi.org/10.3390/ijms19061796) or nerve (https://doi.org/10.1007/s00776-012-0306-9; DOI: 1007/s12010-014-1100-2) regeneration
  2. In line 173, samplesare must be splitted in two words
  3. In line 269, line 290, in Figure 5 and Figure 7 and the rest of the manuscript, μg is not a unit of concentration and must be corrected
  4. Figura 7A is not shown, only 7B
  5. How the presence of morphological alterations regarding cell area, symmetry, width, length and width versus length parameters were calculated should be explained in the materials and methods section 
  6. Lines 374-376 are out of context
  7. In line 380,TiO2NCsshowed must be splitted in two words

Author Response

1- The introduction section is too short and must be improved. For example, examples of biomedical applications of ADSC should be provided such as bladder (https://doi.org/10.1093/rb/rbz049; https://doi.org/10.3390/ijms19061796) or nerve (https://doi.org/10.1007/s00776-012-0306-9; DOI: 1007/s12010-014-1100-2) regeneration

Response:The introduction was improved. We included the references 9-13 with recent studies using ADSC in the biomedical applications (9- DOI: 10.1080/21691401.2019.1594861, 10- DOI: 10.1002/sctm.18-0266, 11- DOI: 10.1093/rb/rbz049, 12- DOI: 10.5001 / omj.2019.97 and 13- DOI: 10.1016 / j.brainres.2020.147025).

We also included recent studies using TiO2 nanoparticles in tissue regeneration (references 2-5, 2 - DOI: 10.1016 / j.ijbiomac.2019.11.246, 3- DOI: 10.1016 / j.btre.2019.e00350, 4- DOI: 10.1016 / j.ijbiomac.2019.07.125 and 5- DOI: 10.1016/j.msec.2019.109770).

2- In line 173, samplesare must be splitted in two words.

Response: The words were corrected.

3- In line 269, line 290, in Figure 5 and Figure 7 and the rest of the manuscript, μg is not a unit of concentration and must be corrected.

Response: The concentration is μg/mL and we updated the manuscript to correct these discrepancies.

4- Figure 7A is not shown, only 7B

Response: We changed the number of figure 7 to clarify the text, now it is figure 8. There is only one figure, then the text was corrected and removed the identification 7A e 7B.

5- How the presence of morphological alterations regarding cell area, symmetry, width, length and width versus length parameters were calculated should be explained in the materials and methods section 

Response: We added the sentence below in the Materials and Methods section after the viability assay methodology, because the analyses of morphological alterations were performed using the Operetta High Content System like for viability assay:

“Morphological alterations regarding cell area, symmetry, width, length and width versus length parameters were calculated using the Harmony Software version 3.5.2 (Perkin Elmer)”.

6- Lines 374-376 are out of context

Response: These lines have been removed from the text.

7- In line 380,TiO2NCsshowed must be splitted in two words

Response: The words were corrected.

Reviewer 3 Report

The topic is not new and lacks innovation as well. This paper is not a high-quality research and cannot make a significant contribution to the field. Therefore, the manuscript cannot be considered as a high quality scientific research paper, and cannot be recommended for publication in present form. 1. The author should rearrange the abstract particularly biological activities. Some abbreviations are not described well. 2. The introduction is too short and need to be elaborated by adding the new knowledge on TiO2 NPs and its impact in biomedical applications. 3. Line 74-77. How the titanium isopropoxide was handled in the room temperature, which need to explained according to chemistry point of view, it produces some white gas. How it opened? 4. Line 180-181 Author mentioned XRD were used to measure the size of the NCs but the results are not shown. Therefore, I suggest author to measure the size of the NCs using the XRD data using the formula described earlier (scherrer equation) with details calculations in supplementary section. 5. Fig.6 is not clear to review the results and the presentation is very poor. 6. I recommend the authors to use the TEM to show the internalization NCs. 7. Overall, this work can be considered for publication after the revision.  

Author Response

1- The author should rearrange the abstract particularly biological activities. Some abbreviations are not described well.

Response: We described the acronym AT-MSCs and NCs in the abstract and we rearranged the abstract taking into account the order of biological tests. Recent studies about AT-MSC and TiO2 in biomedical applications were cited.

2- The introduction is too short and need to be elaborated by adding the new knowledge on TiO2 NPs and its impact in biomedical applications.

Response: The introduction was improved. We included recent studies using TiO2 nanoparticles in the tissue regeneration (references 2-5, 2 - DOI: 10.1016 / j.ijbiomac.2019.11.246, 3- DOI: 10.1016 / j.btre.2019.e00350, 4- DOI: 10.1016 / j.ijbiomac.2019.07.125 and 5- DOI: 10.1016/j.msec.2019.109770).We also showed recent studies using AT-MSC in the biomedical applications (references 9-13, 9- DOI: 10.1080/21691401.2019.1594861, 10- DOI: 10.1002/sctm.18-0266, 11- DOI: 10.1093/rb/rbz049, 12- DOI: 10.5001 / omj.2019.97 and 13- DOI: 10.1016 / j.brainres.2020.147025).

3- Line 74-77. How the titanium isopropoxide was handled in the room temperature, which need to explained according to chemistry point of view, it produces some white gas. How it opened?

Response: This synthesis procedure was based on the procedure published in other articles, with change in reagent concentration [FOOD AND CHEMICAL TOXICOLOGY, v. 112, p. 273-281, 2018; Food and Chemical Toxicology, v. 96, p. 309-319, 2016.] Thus, in an exhaust hood, the reagent was opened, the pipetting process was carried out and the liquid was added to the solution. This procedure is performed on several TiO2 syntheses.

4- Line 180-181 Author mentioned XRD were used to measure the size of the NCs but the results are not shown. Therefore, I suggest author to measure the size of the NCs using the XRD data using the formula described earlier (scherrer equation) with details calculations in supplementary section.

Response: This procedure of determining the size of the crystallite based on the XRD data by the Scherrer equation is well established in the literature, as follows, the procedures. The Scherrer equation: , where,  the mean size of the ordered (crystalline) domains, which may be smaller or equal to the grain size, which may be smaller or equal to the particle size; k is a dimensionless shape factor, with a value close to unity (0.9), s the X-ray wavelength (1.52 A),  is the line broadening at half the maximum intensity (FWHM), after subtracting the instrumental line broadening, in radians. is the Bragg angle. In the samples thermal annealing at 100°C/24h, 500°C/1h, 650°C/1h and 800°C/1h where crystallite sizes corresponding to the predominant phase were calculated.

We added the sentence below in the Materials and Methods section:

“The size of the NCs was estimated based on the Debye-Scherrer equation (Guinier, A. X-ray diffraction in crystals, imperfect crystals, and amorphous bodies. W.H. Freeman: San Francisco, 1963) with 4.4 (anatase), 8.5 (anatase), 32 (rutile) and 44.1 nm (rutile) for thermal annealing at 100°C/24h, 500°C/1h, 650°C/1h and 800°C/1h, respectively.”

5- Fig.6 is not clear to review the results and the presentation is very poor.

Response: The figure 6 wants to illustrate the viability assay results showed in the figure 5. We increased the size of the figure 8 images with PI and Hoeschst staining for shown the cell viability and excluded the others images (bright-field image and Hoechst staining). Hoechst 33342 is a kind of blue-fluorescence dye (excitation/ emission maxima ~350/461 nm), and Propidium iodide (PI), a red-fluorescence dye (excitation/emission maxima ~535/617nm when bound to DNA), is only permeable to dead cells. The majority of the cells showed staining blue because the cell viability for the treatments was more than 80%.

6- I recommend the authors to use the TEM to show the internalization NCs.

Response: This is a very good suggestion, but due to the consequences that the pandemic of the new Coronavirus brought to the routine of our laboratory and the time we have to send the modifications suggested by you, the trial will not be viable.

We added a figure 7A with the Fluorescent imaging of AT-MSCs after 24 hours of treatment with RB NCs in concentrations of 5, 50, 100 and 250 μg (40x).

7- Overall, this work can be considered for publication after the revision.

Response: Thank you so much for your comments and suggestions.   

Round 2

Reviewer 2 Report

The authors have addressed well the reviewers' comments and now the manuscript is ready for publication.

Author Response

Thank you very much

Reviewer 3 Report

This MS can be accepted for publication 

Author Response

Thank you very much